# Learning Nearly Decomposable Value Functions Via Communication Minimization

**Tonghan Wang**[1][*]**, Jianhao Wang**[1][*]**, Chongyi Zheng**[2] **& Chongjie Zhang**[1]

[1]Institute for Interdisciplinary Information Sciences, Tsinghua University, Beijing, China
[2]Turing AI Institute of Nanjing, Nanjing, China
`wangth18@mails.tsinghua.edu.cn, wjh720.eric@gmail.com`
`chongyeezheng@gmail.com, chongjie@tsinghua.edu.cn`

## Abstract

Reinforcement learning encounters major challenges in multi-agent settings, such as scalability and non-stationarity. Recently, value function factorization learning emerges as a promising way to address these challenges in collaborative multi-agent systems. However, existing methods have been focusing on learning fully decentralized value functions, which are not efficient for tasks requiring communication. To address this limitation, this paper presents a novel framework for learning *nearly decomposable Q-functions* (NDQ) via communication minimization, with which agents act on their own most of the time but occasionally send messages to other agents in order for effective coordination. This framework hybridizes value function factorization learning and communication learning by introducing two information-theoretic regularizers. These regularizers are maximizing mutual information between agents' action selection and communication messages while minimizing the entropy of messages between agents. We show how to optimize these regularizers in a way that is easily integrated with existing value function factorization methods such as QMIX. Finally, we demonstrate that, on the StarCraft unit micromanagement benchmark, our framework significantly outperforms baseline methods and allows us to cut off more than $80\%$ of communication without sacrificing the performance. The videos of our experiments are available at `https://sites.google.com/view/ndq`.

## 1 Introduction

Cooperative multi-agent reinforcement learning (MARL) are finding applications in many real-world domains, such as autonomous vehicle teams (Cao et al., 2012), intelligent warehouse systems (Nowé et al., 2012), and sensor networks (Zhang & Lesser, 2011). To help address these problems, recent years have made a great progress in MARL methods (Lowe et al., 2017; Foerster et al., 2018; Rashid et al., 2018; Jaques et al., 2019). Among these successes, the paradigm of centralized training with decentralized execution has attracted much attention for its scalability and ability to deal with non-stationarity.

Value function decomposition methods provide a promising way to exploit such paradigm. They learn a decentralized Q function for each agent and use a mixing network to combine these local Q values into a global action value. In previous works, VDN (Sunehag et al., 2018), QMIX (Rashid et al., 2018), and QTRAN (Son et al., 2019) have progressively enlarged the family of functions that can be represented by the mixing network. Despite their increasing ability in terms of value factorization representation, existing methods have been focusing on learning full decomposition, where each agent acts upon its local observations. However, many multi-agent tasks in the real world are not fully decomposable – agents sometimes require information from other agents in order to effectively coordinate their behaviors. This is because partial observability and stochasticity in a multi-agent environment can exacerbate an agent's uncertainty of other agents' states and actions during decentralized execution, which may result in catastrophic miscoordination.

---

[*]Equal Contribution.

To address this limitation, this paper presents a scalable multi-agent learning framework for learning *nearly decomposable Q-functions* (NDQ) via communication minimization, with which agents act on their own most of the time but occasionally send messages to other agents in order for effective coordination. This framework hybridizes value function factorization learning and communication learning by introducing an information-theoretic regularizer for maximizing mutual information between agents' action selection and communication messages. Messages are parameterized in a stochastic embedding space. To optimize communication, we introduce an additional information-theoretic regularizer to minimize the entropy of messages between agents. With these two regularizers, our framework implicitly learn when, what, and with whom to communicate and also ensure communication to be both *expressive* (i.e., effectively reducing the uncertainty of agents' action-value functions) and *succinct* (i.e., only sending useful and necessary information). To optimize these regularizers, we derive a variational lower bound objective, which is easily integrated with existing value function factorization methods such as QMIX.

We demonstrate the effectiveness of our learning framework on StarCraft II[1] unit micromanagement benchmark used in Foerster et al. (2017; 2018); Rashid et al. (2018); Samvelyan et al. (2019). Empirical results show that NDQ significantly outperforms baseline methods and allows to cut off more than $80\%$ communication without sacrificing the performance. We also observe that agents can effectively learn to coordinate their actions at the cost of sending one or two bits of messages even in complex StarCraft II tasks.

## 2 BACKGROUND

In our work, we consider a fully cooperative multi-agent task that can be modelled by a Dec-POMDP (Oliehoek et al., 2016) $G = \langle I, S, A, P, R, \Omega, O, n, \gamma \rangle$, where $I \equiv \{1, 2, ..., n\}$ is the finite set of agents. $s \in S$ is the true state of the environment from which each agent $i$ draws an individual partial observation $o_i \in \Omega$ according to the observation function $O(s, i)$. Each agent has an action-observation history $\tau_i \in T \equiv (\Omega \times A)^*$. At each timestep, each agent $i$ selects an action $a_i \in A$, forming a joint action $\boldsymbol{a} \in A^n$, resulting in a shared reward $r = R(s, \boldsymbol{a})$ for each agent and the next state $s'$ according to the transition function $P(s'|s, a)$. The joint policy $\boldsymbol{\pi}$ induces a joint action-value function: $Q_{tot}^{\boldsymbol{\pi}}(\boldsymbol{\tau}, \boldsymbol{a}) = \mathbb{E}_{s_{0:\infty}, \boldsymbol{a}_{0:\infty}}[\sum_{t=0}^{\infty} \gamma^t r_t | s_0 = s, \boldsymbol{a}_0 = \boldsymbol{a}, \boldsymbol{\pi}]$, where $\boldsymbol{\tau}$ is the joint action-observation history and $\gamma \in [0, 1)$ is the discount factor.

Learning the optimal action-value function encounters challenges in multi-agent settings. On the one hand, to properly coordinate actions of agents, learning a centralized action-value function $Q_{tot}$ seems a good choice. However, such a function is difficult to learn when the number of agents is large. On the other hand, directly learning decentralized action-value function $Q_i$ for each agent alleviates the scalability problem (Tan, 1993; Tampuu et al., 2017). Nevertheless, such independent learning method largely neglects interactions among agents, which often results in miscoordination and inferior performance.

In between, value function factorization method provides a promising way to attenuate such dilemma by representing $Q_{tot}$ as a mixing of decentralized $Q_i$ conditioned on local information. Such method has shown their effectiveness on complex task (Samvelyan et al., 2019).

However, current value function factorization methods have been mainly focusing on full decomposition. Such decomposition reduces the complexity of learning $Q_{tot}$ by first learning independent $Q_i$ and putting the burden of coordinating actions on the mixing networks whose input is all $Q_i$'s and output is $Q_{tot}$. For many tasks with partial observability and stochastic dynamics, mixing networks are not sufficient to learn coordinated actions, regardless of how powerful its representation ability is. The reason is that full decomposition cuts off all dependencies among decentralized action-value functions and agents will be uncertain about states and actions of other agents. Such uncertainty will increase as time goes by and can result in severe miscoordination and arbitrarily worse performance during decentralized execution.

---

[1]StarCraft and StarCraft II are trademarks of Blizzard Entertainment[TM].

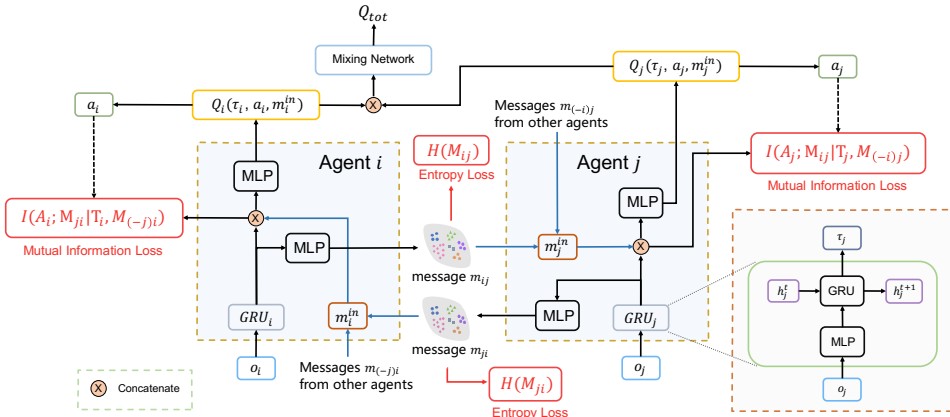

Figure 1: Schematics of our approach. The message encoder generates an embedding distribution that is sampled and concatenated with the current local history to serve as an input to the local action-value function. Local action values are fed into a mixing network to to get an estimation of the global action value.

## 3 METHODOLOGY

In this section, we propose to learn *nearly decomposable Q-functions* (NDQ) via communication minimization, a new framework to overcome the miscoordination issue of full factorization methods.

In our learning framework (Fig. 1), individual action-value functions condition on local action-observation history and, at certain timesteps, messages received from a few other agents. Messages from agent $i$ to agent $j$ are drawn from a multivariate Gaussian distribution whose parameters are given by an encoder $f_m(\tau_i, j; \boldsymbol{\theta}_c)$, where $\tau_i$ is the local observation-action history of agent $i$, and $\boldsymbol{\theta}_c$ are parameters of the encoder $f_m$. Formally, message $m_{ij} \sim \mathcal{N}(f_m(\tau_i, j; \boldsymbol{\theta}_c), \boldsymbol{I})$, where $\boldsymbol{I}$ is an identity matrix. Here we use an identity covariance matrix and the reasons will be discussed in the next section. $m_{(-i)j}$ is used to denote the messages sent to $j$ from agents other than $i$. We learn a nearly decomposable structure via learning minimized communication. We thus expect the communication to have the following properties:

**i) Expressiveness:** The message passed to one agent should effectively reduce the uncertainty in its action-value function.

**ii) Succinctness:** Agents are expected to send messages as short as possible to the agents who need it and only when necessary.

To learn such a communicating strategy, we draw inspiration from variational inference for its proven ability in learning structure from data and endow a stochastic latent message space, which we also refer to as "message embedding". We impose constraints, which will be discussed in detail in the next section, on the latent message embedding to enable an agent to decide locally which bits in a message should be sent according to their utility in terms of helping other agents make decisions. Agent $j$ will receive an input message $m_j^{in}$ that has been selectively cut, on which it conditions the local action-value function $Q_j(\tau_j, a_j, m_j^{in})$. All the individual Q values are then fed into a mixing network such as that used by QMIX (Rashid et al., 2018).

Apart from the constraints on the message embedding, all the components (the individual action-value functions, the message encoder, and the mixing network) are trained in an end-to-end manner by minimizing the TD loss. Thus, our overall objective is to minimize

$$\mathcal{L}(\boldsymbol{\theta}) = \mathcal{L}_{TD}(\boldsymbol{\theta}) + \lambda \mathcal{L}_c(\boldsymbol{\theta}_c), \tag{1}$$

where $\mathcal{L}_{TD}(\boldsymbol{\theta}) = [r + \gamma \max_{\boldsymbol{a'}} Q_{tot}(\boldsymbol{\tau'}, \boldsymbol{a'}; \boldsymbol{\theta}^-) - Q_{tot}(\boldsymbol{\tau}, \boldsymbol{a}; \boldsymbol{\theta})]^2$ ($\boldsymbol{\theta}^-$ are the parameters of a periodically updated target network as in DQN) is the TD loss, $\boldsymbol{\theta}$ are all parameters in the model,

and $\lambda$ is a weighting term. We will discuss how to define and optimize $\mathcal{L}_c(\boldsymbol{\theta}_c)$ to regularize the message embedding in the next section.

### 3.1 Minimized Communication Objective and Variational Bound

Introducing latent variables facilitates the representation of the message, but it does not mean that the messages can reduce uncertainty in the action-value functions of other agents. To make message expressive, we maximize the mutual information between message and agent's action selection. Formally, we maximize $I_{\boldsymbol{\theta}_c}(A_j; M_{ij}|\mathrm{T}_j, M_{(-i)j})$ where $A_j$ is agent $j$'s action selection, $\mathrm{T}_j$ is the random variable of the local action-observation history of agent $j$, $M_{ij}$ and $M_{(-i)j}$ are random variables of $m_{ij}$ and $m_{(-i)j}$. However, if this is the only objective, the encoder can easily learn to cheat by giving messages under different histories representations in different regions in the latent space, rendering cutting off useless messages difficult. A natural constraint to avoid such representations is to minimize the entropy of the messages. Therefore, our objective for optimizing communication of agent $i$ is to maximize:

$$J_c(\boldsymbol{\theta}_c) = \sum_{j=1}^{n} \left[ I_{\boldsymbol{\theta}_c}(A_j; M_{ij}|\mathrm{T}_j, M_{(-i)j}) - \beta H_{\boldsymbol{\theta}_c}(M_{ij}) \right], \tag{2}$$

where $\beta$ is a scaling factor trading expressiveness and succinctness.

This objective is appealing because it agrees exactly with the desiderata that we impose on the message embedding. However, optimizing this objective needs extra efforts because computation involving mutual information is intractable. By introducing a variational approximator, a popular technique from variational toolkit (Alemi et al., 2017), we can derive a lower bound for the mutual information term in Eq. 2 (a detailed derivation can be found in Appendix A):

$$\begin{aligned} &I_{\boldsymbol{\theta}_c}(A_j; M_{ij}|\mathrm{T}_j, M_{(-i)j}) \\ &\geq \mathbb{E}_{\mathbf{T} \sim \mathcal{D}, M_j^{in} \sim f_m(\mathbf{T}, j; \boldsymbol{\theta}_c)} \left[ -\mathcal{CE} \left[ p(A_j|\mathbf{T}) \| q_\xi(A_j|\mathrm{T}_j, M_j^{in}) \right] \right], \end{aligned} \tag{3}$$

where $\mathbf{T} = \langle \mathrm{T}_1, \mathrm{T}_2, \ldots, \mathrm{T}_n \rangle$ is the joint local history sampled from the replay buffer $\mathcal{D}$, $q_\xi(A_j|\mathrm{T}_j, M_j^{in})$ is the variational posterior estimator with parameters $\xi$, and $\mathcal{CE}$ is the cross entropy operator. We share $\xi$ among agents to accelerate learning.

Next we discuss how to minimize the term $H_{\boldsymbol{\theta}_c}(M_{ij})$. Directly minimizing this can cause the variances of the Gaussian distributions to collapse to 0. To deal with this numeric issue, we use the unit covariance matrix and try to minimize $H(M_{ij}) - H(M_{ij}|\mathrm{T}_i)$ instead. This is equivalent to minimizing $H(M_{ij})$ because $H(M_{ij}|\mathrm{T}_i)$ is the entropy of a multivariate Gaussian random variable and thus is a constant $\log(\det(2\pi e \boldsymbol{\Sigma}))/2$, where $\boldsymbol{\Sigma}$ is a unit matrix in our formulation). Then we have:

$$H(M_{ij}) - H(M_{ij}|\mathrm{T}_i) = \int p(m_{ij}|\tau_i) p(\tau_i) \log \frac{p(m_{ij}|\tau_i)}{p(m_{ij})} dm_{ij} d\tau_i. \tag{4}$$

We use a similar technique as for the mutual information term by introducing an distribution $r(m_{ij})$ to get a upper bound of Eq. 4:

$$\begin{aligned} H(M_{ij}) - H(M_{ij}|\mathrm{T}_i) &\leq \int p(m_{ij}|\tau_i) p(\tau_i) \log \frac{p(m_{ij}|\tau_i)}{r(m_{ij})} dm_{ij} d\tau_i \\ &= \mathbb{E}_{\mathrm{T}_i \sim D} \left[ D_{\mathrm{KL}}(p(M_{ij}|\mathrm{T}_i) \| r(M_{ij})) \right]. \end{aligned} \tag{5}$$

This bound holds for any distribution $r(M_{ij})$. To facilitate cutting off messages, we use unit Gaussian distribution $\mathcal{N}(0, \boldsymbol{I})$. Combining Eq. 3 and 5, we get a tractable variational lower bound of our objective in Eq. 2:

$$J_c(\boldsymbol{\theta}_c) \geq \mathbb{E}_{\mathbf{T} \sim \mathcal{D}, M_j^{in} \sim f_m(\mathbf{T}, j; \boldsymbol{\theta}_c)} \left[ -\mathcal{CE} \left[ p(A_j|\mathbf{T}) \| q_\xi(A_j|\mathrm{T}_j, M_j^{in}) \right] - \beta D_{\mathrm{KL}}(p(M_{ij}|\mathrm{T}_i) \| r(M_{ij})) \right]. \tag{6}$$

We optimize this bound to generate an expressive and succinct message embedding. Specifically, we minimize:

$$\mathcal{L}_c(\boldsymbol{\theta}_c) = \mathbb{E}_{\mathbf{T} \sim \mathcal{D}, M_j^{in} \sim f_m(\mathbf{T}, j; \boldsymbol{\theta}_c)} \left[ \mathcal{CE} \left[ p(A_j|\mathbf{T}) \| q_\xi(A_j|\mathrm{T}_j, M_j^{in}) \right] + \beta D_{\mathrm{KL}}(p(M_{ij}|\mathrm{T}_i) \| r(M_{ij})) \right]. \tag{7}$$

Intuitively, the first term, which we call the *expressiveness loss*, ensures that communication aims to reduce the uncertainty in action-value functions of other agents. The second term, called the *succinctness loss*, forces messages to get close to the unit Gaussian distribution. Since we set the covariances of the latent message variables to the unit matrix, this term actually pushes the means of the message distributions to the origin of the latent space. Using these two losses leads to an embedding space where useless messages distribute near the origin, while messages that contain important information for the decision-making processes of other agents occupy other spaces.

Note that the loss shown in Eq. 7 is used to update the parameters in the message encoder. In the meantime, all components (the individual action-value functions, the message encoder, and the mixing network) are trained in an end-to-end manner. Thus, the message encoder is updated by two gradients: the gradient induced by $\mathcal{L}_c(\boldsymbol{\theta}_c)$ and the gradient associated with the TD loss $\mathcal{L}_{TD}(\boldsymbol{\theta})$.

## 3.2 Cutting Off Messages

Our objective pushes messages which can not reduce the uncertainties in action-value functions of other agents close to the origin of the latent message space. This naturally gives us a hint on how to drop meaningless messages – we can order the message distributions according to their means and drop accordingly. Note that since we use a unit covariance matrix for the latent message distribution, bits in a message are independent. Thus, we can make decisions in a bit-by-bit fashion and send messages with various lengths. In this way, our method learns not only when and who (agent $i$ does not communicate with agent $j$ when all bits of $m_{ij}$ are dropped) to communicate, but also what to communicate (how many bits are sent and their values). More details are discussed in Appendix B.

Our framework adopts the centralized training with decentralized execution paradigm. During centralized training, we assume the learning algorithm has access to all agents' individual observation-action histories and the global state $s$. During execution, agents communicate and act in a decentralized fashion based on the learned message encoder and action-value functions.

## 4 Related Works

Deep multi-agent reinforcement learning has witnessed vigorous progress in recent years. COMA (Foerster et al., 2018), MADDPG (Lowe et al., 2017), and PR2 (Wen et al., 2019) explores multi-agent policy gradients and respectively address the problem of credit assignment, learning in mixed environments and recursive reasoning. Another line of research focuses on value-based multi-agent RL, among which value-function factorization is the most popular method. Three representative examples: VDN (Sunehag et al., 2018), QMIX (Rashid et al., 2018), and QTRAN (Son et al., 2019) gradually increase the representation ability of the mixing network. In particular, QMIX (Rashid et al., 2018) stands out as a scalable and robust algorithm and achieves state-of-the-art results on StarCraft unit micromanagement benchmark (Samvelyan et al., 2019).

Communication is a hot topic in multi-agent reinforcement learning. End-to-end learning with differentiable communication channel is a popular approach now. Sukhbaatar et al. (2016); Hoshen (2017); Jiang & Lu (2018); Singh et al. (2019); Das et al. (2019) focus on learning decentralized communication protocol and address the problem of when and who to communicate. Foerster et al. (2016); Das et al. (2017); Lazaridou et al. (2017); Mordatch & Abbeel (2018) study the emergence of natural language in the context of multi-agent learning. IC3Net (Singh et al., 2019) learns gate to control the agents to only communicate with their teammates in mixed multi-agent environment. Zhang & Lesser (2013); Kim et al. (2019) study action coordination under limited communication channel and thus are related to our works. The difference lies in that they do not explicitly minimize communication. Social influence (Jaques et al., 2019) and InfoBot (Goyal et al., 2019) penalize message that has no influence on policies of other agents.

Work that is most related to this paper is TarMAC (Das et al., 2019), where attention mechanism is used to differentiate the importance of incoming messages. In comparison, we use variation inference to decide the content of messages and whether a message should be sent under the guidance of global reward signals. We compare our method with TarMAC and a baseline combining TarMAC and QMIX in our experiments. Related works on the task of StarCraft II unit micromanagement are discussed in Appendix C.2.

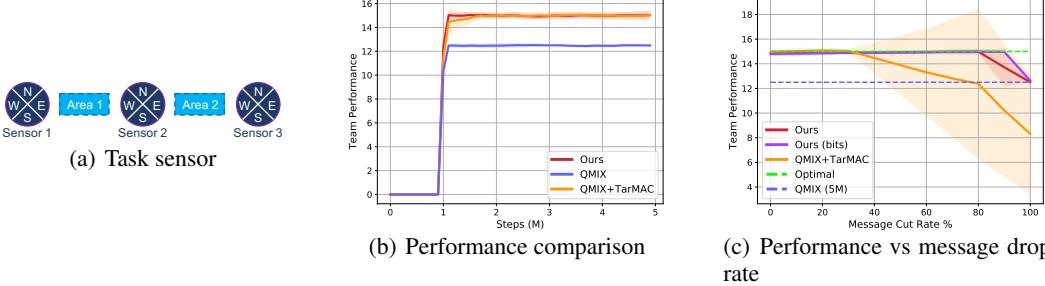

(a) Task sensor

(b) Performance comparison

(c) Performance vs message drop rate

Figure 2: (a) Task **sensor**; (b) Performance comparison on *sensor*; (c) Performance comparison when different percentages of messages are dropped. We measure the drop rate of our method in two ways: count by the number of messages (NDQ) or count by the number of bits (NDQ (bits)). QMIX (5M) is the performance of QMIX after training for 5 million time steps.

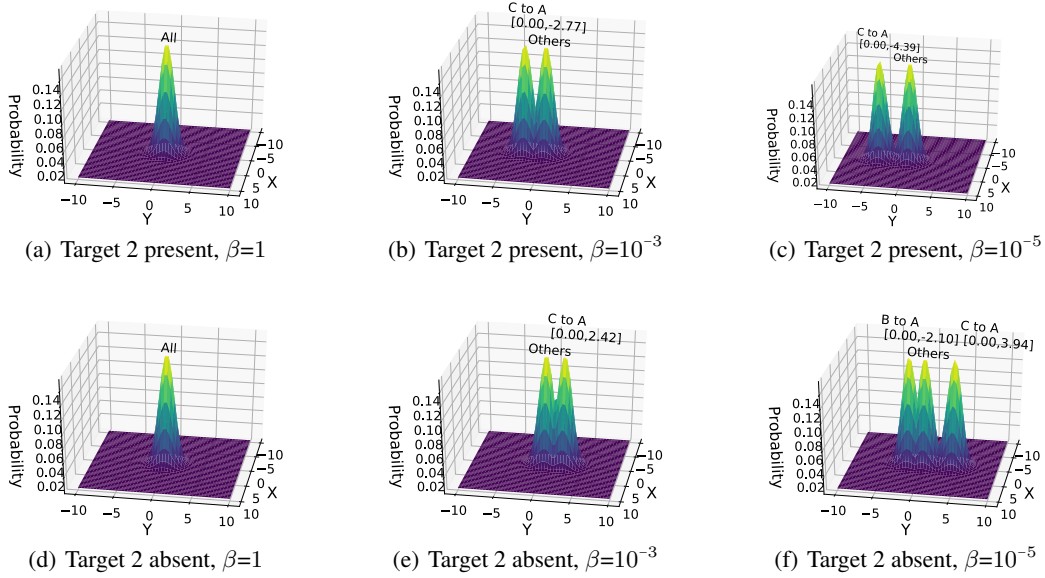

(a) Target 2 present, $\beta=1$

(b) Target 2 present, $\beta=10^{-3}$

(c) Target 2 present, $\beta=10^{-5}$

(d) Target 2 absent, $\beta=1$

(e) Target 2 absent, $\beta=10^{-3}$

(f) Target 2 absent, $\beta=10^{-5}$

Figure 3: Message distributions learned by our method on *sensor* under different values of $\beta$. (Messages are cut by bit, if $\mu < 2.0$). A mean of 0 means that the corresponding bit is below the cutting threshold and is not sent. When $\beta = 10^{-3}$, NDQ learns the minimized communication strategy that is effective.

## 5 EXPERIMENTAL RESULTS

In this section, we show our experiments to answer the following questions: (i) Is the miscoordination problem of full value function factorization methods widespread? (ii) Can our method learn the minimized communication protocol required by a task? (iii) Can the learned message distributions reduce uncertainties in value functions of other agents? (iv) How does our method differ from communication with attention mechanism? (v) How does $\beta$ influence the communication protocol? We will first show three simple examples to clarify our idea from different perspectives and then provide performance analysis on StarCraftII unit micromanagement benchmark. For evaluation, all experiments are carried out with 5 random seeds and results are shown with a 95% confidence interval. Details of the NDQ network architecture are given in Appendix B.2. Videos of our experiments on StarCraft II are available online[2].

---

[2]https://sites.google.com/view/ndq

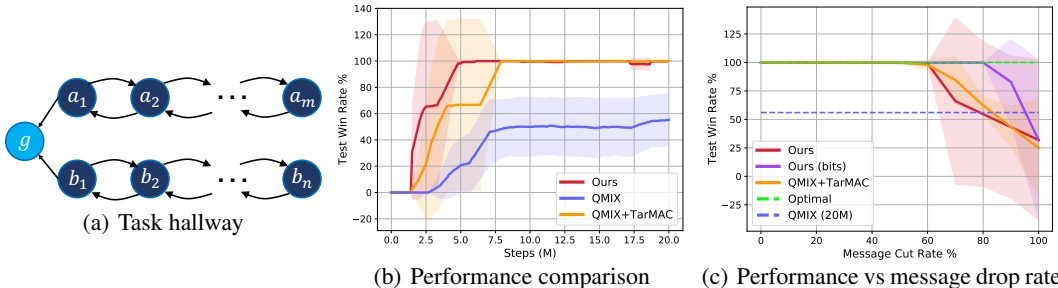

(a) Task hallway     (b) Performance comparison     (c) Performance vs message drop rate

Figure 4: Results on **hallway**. (a, b) Task *hallway* and performance comparison. (c) Similar to Fig. 2(c), we show performance comparison when different percentages of messages are dropped.

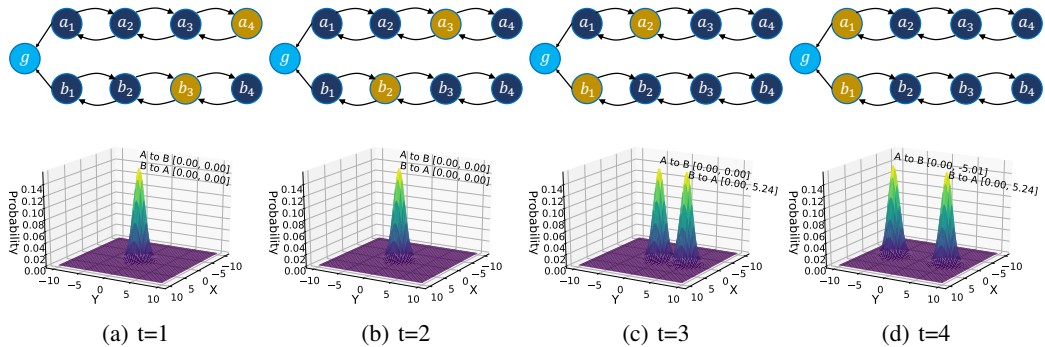

(a) t=1     (b) t=2     (c) t=3     (d) t=4

Figure 5: Message embedding representations learned by our method on *hallway*. A mean of 0 means that the corresponding bit is below the cutting threshold ($\mu=3$) and is not sent.

We compare NDQ with the following baselines: (i) QMIX (Rashid et al., 2018); (ii) TarMAC (Das et al., 2019). QMIX and TarMAC are state-of-the-art full value function factorization and attentional communication methods, respectively. (iii) QMIX+TarMAC. We introduce the attentional communication mechanism into the value function factorization paradigm by integrating the communication component of TarMAC into QMIX.

## 5.1 DIDACTIC EXAMPLES

We first demonstrate our idea on three didactic examples: **sensor**, **hallway**, and **independent search**.

Sensor network is a frequently used testbed in multi-agent learning field (Kumar et al., 2011; Zhang & Lesser, 2011). We use a 3-chain sensor configuration in the task **sensor** (Fig. 2(a)). Each sensor is controlled by one agent, and they are rewarded for successfully locating targets, which requires two sensors to scan the same area simultaneously when the target appears. At each timestep, target 1 appears in area 1 with possibility 1, and locating it induces a team reward of 20; target 2 appears with probability 0.5 in area 2, and agents are rewarded 30 for locating it. Agents can observe whether a target is present in nearby areas and need to choose one of the five actions: scanning `north`, `east`, `south`, `west`, and `noop`. Every scan induces a cost of -5.

In the optimal policy, when target 2 appears, sensor 1 should turn itself off while sensors 2 and 3 are expected to scan area 2 to get the reward. And when target 2 is absent, sensors 1 and 2 need to cooperatively scan area 1 while sensor 3 takes `noop`.

*Sensor* is representative of a class of tasks where the uncertainties about the true states cause policies learned by full value function factorization method to be sub-optimal – sensor 1 has to know whether the target is present in area 2 to make a decision. However, the mixing network of QMIX cannot provide such information. As a result, QMIX converges to a sub-optimal policy, which gets a team reward of 12.5 a step on average (see Fig. 2(b)).

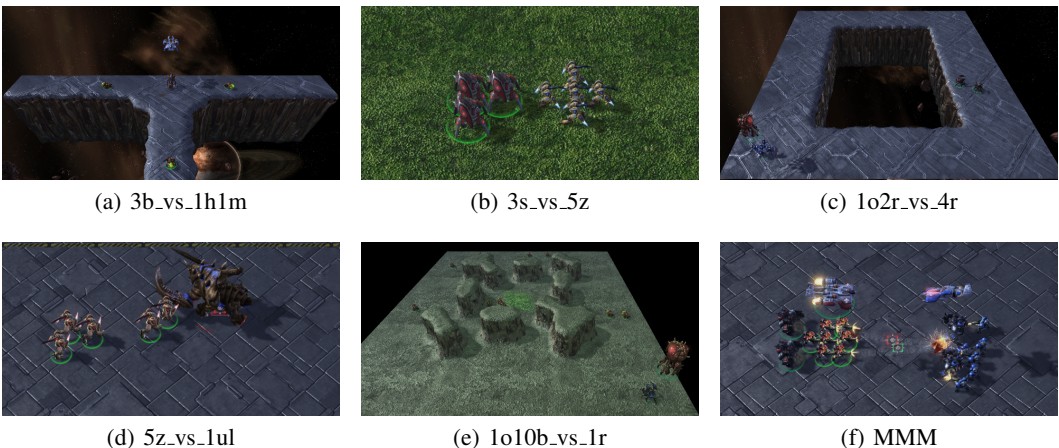

(a) 3b_vs_1h1m  (b) 3s_vs_5z  (c) 1o2r_vs_4r

(d) 5z_vs_1ul  (e) 1o10b_vs_1r  (f) MMM

Figure 6: Snapshots of the StarCraft II scenarios that we consider.

We are particularly interested in whether our method can learn the minimized communication strategy. Fig. 3 shows the latent message space learned by NDQ. When $\beta = 10^{-3}$, agent 3 learns to send a bit to tell agent 1 whether target 2 appears. In the meantime, the latent message distribution between any other pair of agents is close to the standard Gaussian distribution and thus is dropped. This result indicates that NDQ has discovered the minimized conditional graph and can explain why our method can still perform optimally when $80\%$ of the messages are cut off (Fig. 2(c)). When $\beta$ becomes too large (1.0), all the message bits are pushed below the cutting threshold (Fig. 3(a) and 3(d)). When $\beta$ is too small ($10^{-5}$), NDQ pays more attention on reducing uncertainties in Q-functions rather than compressing messages. Correspondingly, both agent 3 and agent 2 send a message to agent 1 (Fig. 3(c) and 3(f)), which is a redundant communication strategy.

The second example, **hallway** (Fig. 4(a)), is a Dec-POMDP with two agents randomly starting at states $a_1$ to $a_m$ and $b_1$ to $b_n$, respectively. Agents can observe their position and choose to move left, move right, or keep still at each timestep. Agents will win and get a reward of 10 if they arrive at state $g$ simultaneously. Otherwise, if any agent arrives at $g$ earlier than the other, the team will not be rewarded, and the next episode will begin. The horizon is set to $\max(m, n) + 10$ to avoid an infinite loop.

*Hallway* aims to show that the miscoordination problem of full factorization methods can be severe in multi-step scenarios. We set $m$ and $n$ to 4 and show comparison of performance in Fig. 4(b). The miscoordination problem causes QMIX to lose about half of the games. We are again particularly interested in the message embedding representations learned by NDQ. We show an episode in Fig. 5. Two agents begin at $a_4$ and $b_3$, respectively. They first move left silently ($t = 1$ and $t = 2$) until agent B arrives at $b_1$. On arriving $b_1$, it sends a bit whose value is 5.24 to A. After sending this bit, B stays at $b_1$ and sends this message repeatedly until it receives a bit from A indicating that A has arrived at $a_1$. They then move left together and win. This is the minimized communication strategy. Taking advantage of this strategy, NDQ can still win in 100% of episodes when 80% of the communicating bits are dropped (Fig. 4(c)).

The third task, **independent search**, aims to demonstrate that NDQ can learn not to communicate in scenarios where agents are independent. Task description and results analysis are deferred to Appendix C.1.

## 5.2 MAXIMUM VALUE FUNCTION FACTORIZATION IN STARCRAFT II

To demonstrate that the miscoordination problem of full decomposition methods is widespread in multi-agent learning, we apply our method and baselines to the StarCraft II micromanagement benchmark introduced by Samvelyan et al. (2019), which is described in detail in Appendix C.2. We further increase the difficulty of action coordination by i) reducing the sight range of agents from 9 to 2; ii) introducing challenging maps with complex terrain or highly random spawning po-

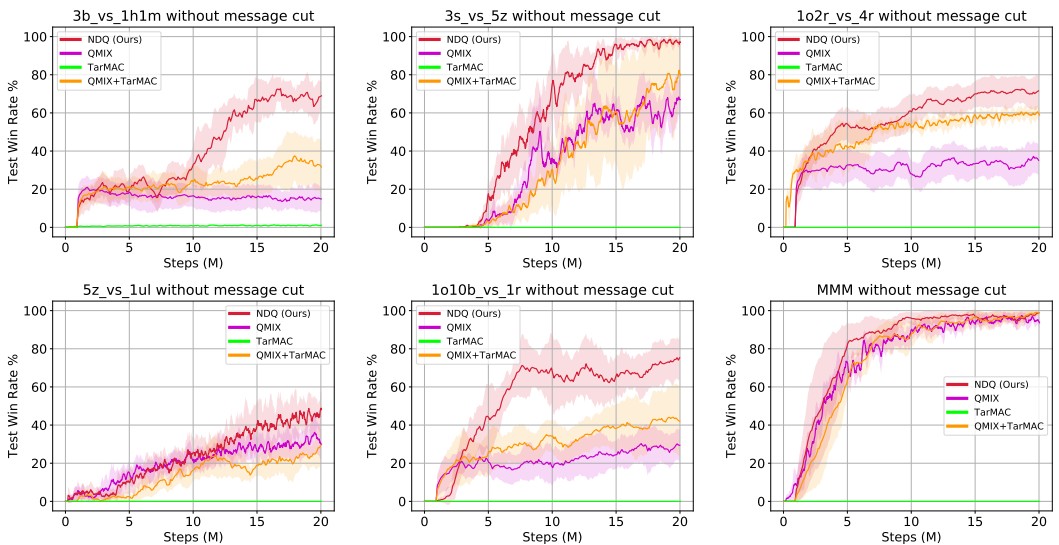

Figure 7: Learning curves of our method and baselines when no message is cut for NDQ and QMIX+TarMAC.

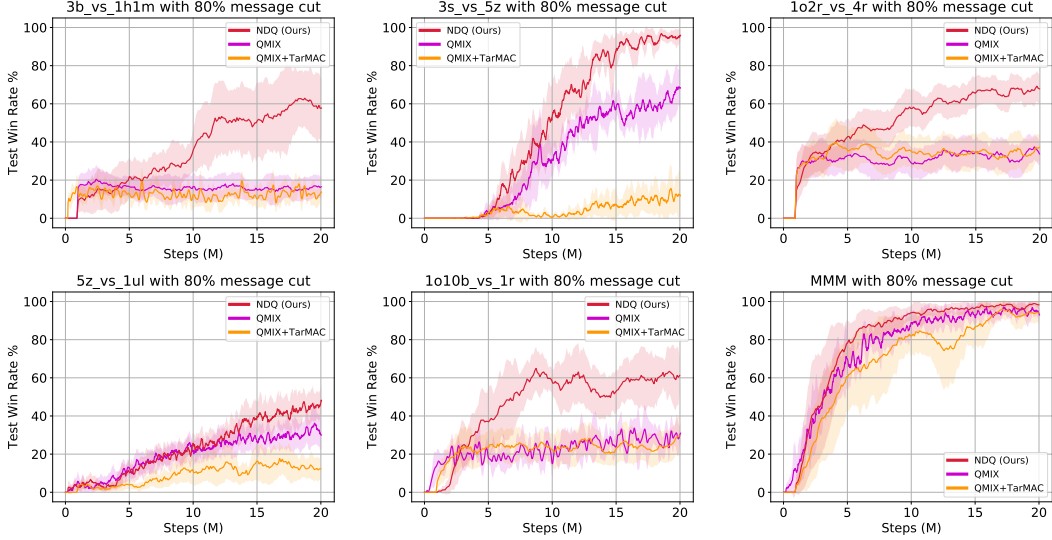

Figure 8: Performance of our method and QMIX+TarMAC when $80\%$ of messages are cut off. We also plot the learning curves of QMIX for comparison.

sitions for units. We test our method on the six maps shown in Fig. 6. Detailed descriptions of these scenarios are provided in Appendix C.2.

We use the same hyper-parameter setting for NDQ on all maps: $\beta$ is set to $10^{-5}$, $\lambda$ is set to $0.1$, and the length of message $m_{ij}$ is set to 3. For evaluation, we pause training every $100k$ environment steps and run 48 testing episodes. Other hyper-parameters for NDQ are described in Appendix B.2.

### 5.2.1 PERFORMANCE COMPARISON

We show the performance of our method and baselines when no message is cut in Fig. 7. The superior performance of NDQ against QMIX demonstrates that the miscoordination problem of full factorization methods is widespread, especially in scenarios with high stochasticity, such as 1o2r_vs_4r, 3b_vs_1h1m, and 1o10b_vs_1r. Notably, our method also outperforms the attentional

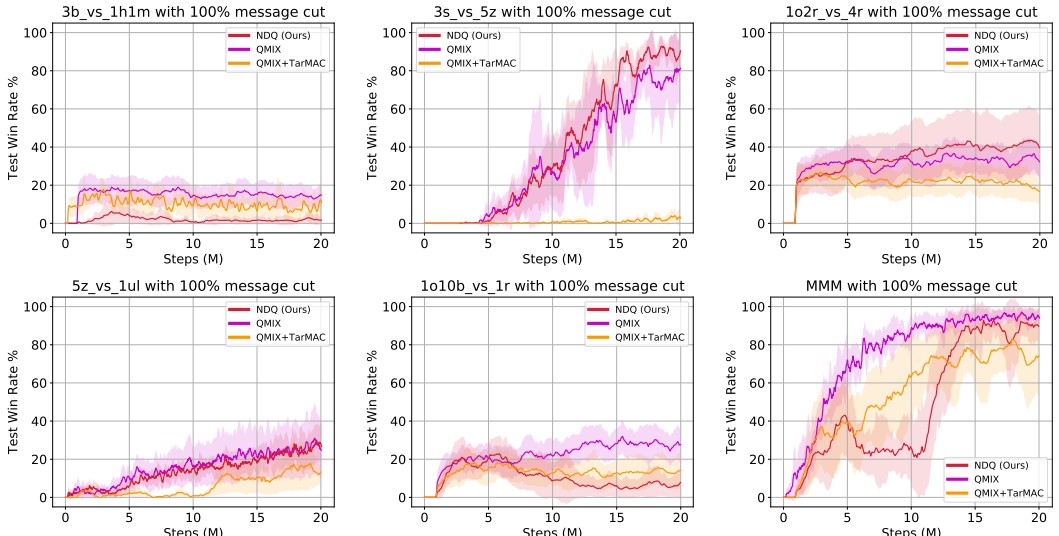

Figure 9: Performance of our method and QMIX+TarMAC when $100\%$ messages are cut off. We also plot the learning curves of QMIX for comparison.

communication mechanism (QMIX+TarMAC) by a large margin. Since agents communicate in both of these two methods and the same TD error is used, these results highlight the role of the constraints that we impose on our message embedding. TarMAC struggles in all the scenarios. We believe that this is because it does not deal with the issue of reward assignment.

### 5.2.2 MESSAGE CUT OFF

To demonstrate that our method can learn nearly decomposable Q-functions in complex tasks, we cut off $80\%$ of messages according to the means of distributions when testing and show the results in Fig. 8. The results indicate that we can omit more than $80\%$ of communication without significantly affecting performance. For comparison, we cut off messages in QMIX+TarMAC whose weights are $80\%$ smallest and find that its performance drops significantly (Fig. 8). These results indicate that our method is more robust in terms of message cutting off compared to the attentional communication methods.

We further drop all the messages and show the developments of testing performance in Fig. 9. As expected, the win rates of NDQ decrease dramatically, proving that the superiority of our method when $80\%$ of messages are dropped comes from expressive and succinct communication protocols.

## 6 CLOSING REMARKS

In this paper, we presented a novel multi-agent learning framework within the paradigm of centralized training with decentralized execution. This framework fuses value function factorization learning and communication learning and efficiently learns nearly decomposable value functions for agents to act most of the time independently and communicate when it is necessary for coordination. We introduce two information-theoretical regularizers to minimize overall communication while maximizing the message information for coordination. Empirical results in challenging Star-Craft II tasks show that our method significantly outperforms baseline methods and allows us to reduce communication by more than $80\%$ without sacrificing the performance. We also observe that nearly minimal messages (e.g., with one or two bits) are learned to communicate between agents in order to ensure effective coordination.

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

APPENDIX

## A  VARIATIONAL BOUND ON MUTUAL INFORMATION

In order to enable messages to effectively reduce the uncertainties in action-value functions of other agents, we propose to maximize the mutual information between $A_j$ and $M_{ij}$. We borrow ideas from the variational inference literature and derive a lower bound of this mutual information regularizer.

**Theorem 1.** *A lower bound of mutual information $I_{\boldsymbol{\theta}_c}(A_j; M_{ij}|\mathrm{T}_j, M_{(-i)j})$ is*

$$\mathbb{E}_{\mathbf{T}\sim\mathcal{D}, M_j^{in}\sim f_m(\mathbf{T}, j;\boldsymbol{\theta}_c)}\left[-\mathcal{CE}\left[p(A_j|\mathbf{T})\|q_\xi(A_j|\mathrm{T}_j, M_j^{in})\right]\right], \tag{8}$$

*where $\mathrm{T}_j$ is the local action-observation history of agent $j$, and $\mathbf{T} = \langle \mathrm{T}_1, \mathrm{T}_2, \ldots, \mathrm{T}_n\rangle$ is the joint local history sampled from the replay buffer $\mathcal{D}$, $q_\xi(A_j|\mathrm{T}_j, M_j^{in})$ is the variational posterior estimator with parameters $\xi$.*

*Proof.*

$$I_{\boldsymbol{\theta}_c}(A_j; M_{ij}|T_j, M_{(-i)j}) \tag{9}$$

$$= \int p(a_j, \tau_j, m_j^{in}) \log \frac{p(a_j, m_{ij}|\tau_j, m_{(-i)j})}{p(a_j|\tau_j, m_{(-i)j})p(m_{ij}|\tau_j, m_{(-i)j})} da_j d\tau_j dm_j^{in} \tag{10}$$

$$= \int p(a_j, \tau_j, m_j^{in}) \log \frac{p(a_j|\tau_j, m_j^{in})}{p(a_j|\tau_j, m_{(-i)j})} da_j d\tau_j dm_j^{in}, \tag{11}$$

where $p(a_j|\tau_j, m_j^{in})$ is determined by the message encoder $f_m$ and Markov Chain:

$$p(a_j|\tau_j, m_j^{in}) \tag{12}$$

$$= \int p(\tau_{-j}, a_j|\tau_j, m_j^{in})d\tau_{-j} \tag{13}$$

$$= \int p(\tau_{-j}|\tau_j, m_j^{in})p(a_j|\boldsymbol{\tau})d\tau_{-j} \quad \left(\text{According to } \left[a_j \perp m_j^{in}|\boldsymbol{\tau}\right]\right) \tag{14}$$

$$= \int \frac{p(\boldsymbol{\tau})p(m_j^{in}|\boldsymbol{\tau})p(a_j|\boldsymbol{\tau})}{p(\tau_j, m_j^{in})}d\tau_{-j}. \tag{15}$$

We introduce $q_\xi(a_j|\tau_j, m_j^{in})$ as a variational approximation to $p(a_j|\tau_j, m_j^{in})$. Since

$$D_{\mathrm{KL}}(p(a_j|\tau_j, m_j^{in})\|q_\xi(a_j|\tau_j, m_j^{in}) \geq 0, \tag{16}$$

we have

$$\int p(a_j|\tau_j, m_j^{in}) \log p(a_j|\tau_j, m_j^{in})da_j \tag{17}$$

$$\geq \int p(a_j|\tau_j, m_j^{in}) \log q_\xi(a_j|\tau_j, m_j^{in})da_j. \tag{18}$$

Thus, for the mutual information term:

$$I_{\boldsymbol{\theta}_c}(A_j; M_{ij}|T_j, M_{(-i)j}) \tag{19}$$

$$\geq \int p(a_j, \tau_j, m_j^{in}) \log \frac{q_\xi(a_j|\tau_j, m_j^{in})}{p(a_j|\tau_j, m_{(-i)j})} da_j d\tau_j dm_j^{in} \tag{20}$$

$$= \int p(a_j, \tau_j, m_j^{in}) \log q_\xi(a_j|\tau_j, m_j^{in})da_j d\tau_j dm_j^{in} \tag{21}$$

$$- \int p(a_j, \tau_j, m_j^{in}) \log p(a_j|\tau_j, m_{(-i)j})da_j d\tau_j dm_j^{in} \tag{22}$$

$$= \int p(\boldsymbol{\tau})p(m_j^{in}|\boldsymbol{\tau})p(a_j|\boldsymbol{\tau}, m_j^{in}) \log q_\xi(a_j|\tau_j, m_j^{in})da_j d\boldsymbol{\tau} dm_j^{in} \tag{23}$$

$$- \int p(a_j, \tau_j, m_{(-i)j}) \log p(a_j|\tau_j, m_{(-i)j}) da_j d\tau_j dm_{(-i)j} \tag{24}$$

$$= \int p(\boldsymbol{\tau})p(m_j^{in}|\boldsymbol{\tau})p(a_j|\boldsymbol{\tau}) \log q_\xi(a_j|\tau_j, m_j^{in}) da_j d\boldsymbol{\tau} dm_j^{in} \quad \left(\text{According to } [a_j \perp m_j^{in}|\boldsymbol{\tau}]\right) \tag{25}$$

$$+ H_{\boldsymbol{\theta}_c}(A_j|\mathrm{T}_j, M_{(-i)j}) \tag{26}$$

$$= \mathbb{E}_{\mathbf{T}\sim\mathcal{D}, M_j^{in}\sim f_m(\mathbf{T},j;\boldsymbol{\theta}_c)} \left[\int p(a_j|\mathbf{T}) \log q_\xi(a_j|\mathrm{T}_j, M_j^{in}) da_j\right] \tag{27}$$

$$+ H_{\boldsymbol{\theta}_c}(A_j|\mathrm{T}_j, M_{(-i)j}) \tag{28}$$

$$= \mathbb{E}_{\mathbf{T}\sim\mathcal{D}, M_j^{in}\sim f_m(\mathbf{T},j;\boldsymbol{\theta}_c)} \left[-\mathcal{CE}\left[p(A_j|\mathbf{T})\|q_\xi(A_j|\mathrm{T}_j, M_j^{in})\right]\right] \tag{29}$$

$$+ H_{\boldsymbol{\theta}_c}(A_j|\mathrm{T}_j, M_{(-i)j}). \tag{30}$$

Because $H_{\boldsymbol{\theta}_c}(A_j|\mathrm{T}_j, M_{(-i)j}) \geq 0$, we get the lower bound in Theorem 1.

$\square$

## B    Implementation Details

### B.1    Details of Message Dropping

In our methods, not only the number of messages but also the length of messages are minimized. In other words, we send messages with varying lengths. However, messages are feed into an action-value function approximator at the recipient side, which requires inputs to have the same length. To solve this problem, we send masks indicating which bits are dropped along with the messages. To save channel bandwidth, masks are regarded as binary numbers, so each of them only consumes a negligible $\log$-scale space compared to the length of messages. For the unsent bits, we fill in 0s before feeding the messages into the local utility functions.

### B.2    Network Architecture, Hyperparameters, and Infrastructure

We base our implementation on the PyMARL framework (Samvelyan et al., 2019) and use its default network structure and hyper-parameter setting for QMIX. For the message encoder, we use a fully connected network with one 64-dimensional hidden layer and ReLU activation. For the posterior estimator $q_\xi$, we adopt a fully connected network with two 20-dimensional hidden layers with ReLU activation. We train our models on NVIDIA RTX 2080Ti GPUs using experience sampled from 16 parallel environments. To benchmark NDQ, we train all algorithms for 20 million time steps on each StarCraft II unit micromanagement task and use the default hyper-parameter settings for baselines.

## C    Experimental Results

### C.1    Didactic example: independent search

In **independent search**, two agents are finding landmarks in two independent $5 \times 5$ rooms for 100 time steps (see Fig. 10). An agent is rewarded 1 when it is on the landmark in its room.

Independent search is an example where agents are totally independent. This task aims to demonstrate that our method can learn not to communicate in independent scenarios. We show team performance in Table 1. NDQ can achieve the optimal performance when agents do not communicate with each other.

Table 1: The average team reward gained in an episode on the task *independent-search*.

|  | Ours | QMIX | TarMAC | TarMAC + QMIX |
|---|---|---|---|---|
| No message is cut | 96.0 | 96.0 | 96.0 | 96.0 |
| 100% messages are cut | 96.0 | — | — | 96.0 |

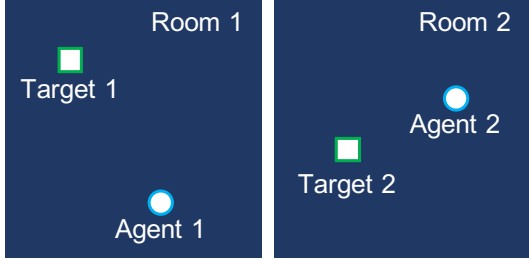

Figure 10: Task **Independent-search**. Two agents are both reward- and transition-independent.

## C.2   STARCRAFT II

StarCraft unit micromanagement has attracted lots of research interests for its high degree of control complexity and environmental stochasticity. Usunier et al. (2017) and Peng et al. (2017) study this problem from a centralized perspective. In order to facilitate decentralized control, we use the setup introduced by Samvelyan et al. (2019).

We first describe the scenarios that we consider in detail. We consider combat scenarios where the enemy units are controlled by StarCraft II built-in AI (difficulty level is set to `medium`), and each of the ally units is controlled by a learning agent. The units of the two groups can be asymmetric, and the initial placement is random. At each time step, each agent chooses one action from the discrete action space consisting of the following actions: `noop`, `move[direction]`, `attack[enemy_id]`, and `stop`. Under the control of these actions, agents move and attack in a continuous map. A global reward that is equal to the total damage dealt on the enemy units is given at each timestep. Killing each enemy unit and winning a combat induce extra bonuses of 10 and 200, respectively.

**3b_vs_1h1m**: 3 Banelings try to kill a Hydralisk assisted by a Medivac. 3 Banelings together can just blow up the Hydralisk. Therefore, they should not give the Hydralisk rest time during which the Medivac can restore its health. Banelings have to attack at the same time to get the winning reward. This scenario is designed to test whether our method can learn a communication protocol to coordinate actions.

**3s_vs_5z**: 3 Stalkers encounter 5 Zealots on a map. Zealots can cause high damage but are much slower so that Stalkers have to take advantage of a technique called *kiting* – Stalkers should alternatively attack the Zealots and flee for a distance.

**1o2r_vs_4r**: An Overseer has found 4 Reapers. Its ally units, 2 Roaches, need to get there and kill the Reapers to win. At the beginning of an episode, the Overseer and Reapers spawn at a random point on the map while the Roaches are initialized at another random point. Given that only the Overseer knows the position of the enemy, a learning algorithm has to learn to deliver this message to the Roaches to effectively win the combat.

**5z_vs_1ul**: 5 Zealots try to kill a powerful Ultralisk. A sophisticated micro-trick demanding right positioning and attack timing has to be learned to win.

**MMM**: Symmetric teams consisting of 7 Marines, 2 Marauders, and 1 Medivac spawn at two fixed points, and the enemy team is tasked to attack the ally team. To win the battle, agents have to learn to communicate their health to the Medivac.

**1o10b_vs_1r**: In a map full of cliffs, an Overseer detects a Roach. The teammates of the Overseer, 10 Banelings, need to kill this Roach to get the winning reward. The Overseer and the Roach spawn at a random point while the Banelings spawn randomly on the map. In the minimized communication strategy, the Banelings can keep silent, and the Overseer needs to encode its position and send it to the Banelings. We use this task to test the performance of our method in complex scenarios.

