# OpenReview forum: "Learning Nearly Decomposable Value Functions Via Communication Minimization"
_ICLR.cc/2020/Conference — Accept (Poster)_

### Official Review · AnonReviewer2 · 2019-10-24
**Official Blind Review #2**

**Rating:** 3

**Review:**

The authors propose a framework for combining value function factorization and communication learning in a multi-agent setting by introducing two regularizers, one for maximizing mutual information between decentralized Q functions and communication messages and the other for minimizing the entropy of messages between agents. The authors also discuss a method for dropping non-informative messages. They illustrate their approach on sensor and hallway tasks and evaluate their method on the decentralized StarCraft II benchmark. The paper addresses an interesting problem, and the authors show that their approach gives good performance compared to alternative approaches even when a large percentage of communication is cut off between the agents.

Questions/Comments:
- Implementation details (e.g., hyper-parameters, model size) are missing from the paper.
- The results are average over only 3 seeds, is this enough to compare different algorithms?
- How should beta should be determined?
- The authors present results when 80% of messages are cut off. What is the performance of the model when all communication is cut off for comparison?
- How does the approach work in competitive environments?
- The experimental results section is not well organized. The authors mention five question on page 6, but it is not very clear with examples/set of experiments address which question.
- There are many spelling/grammar errors in the paper.

**Experience Assessment:**

I do not know much about this area.

**Review Assessment: Checking Correctness Of Derivations And Theory:**

I assessed the sensibility of the derivations and theory.

**Review Assessment: Checking Correctness Of Experiments:**

I assessed the sensibility of the experiments.

**Review Assessment: Thoroughness In Paper Reading:**

I read the paper at least twice and used my best judgement in assessing the paper.

---

> ### Author Response · Authors · 2019-11-11
> **Thanks for your comments. We have run all the SC2 experiments with more random seeds. Here we provide explanations to clarify your questions.**
>
> Thanks for your comments. Here we provide explanations to clarify your questions. In addition, please feel free to refer to our response to reviewer #1, which summarizes novelties of our paper.
>
> Q: Implementation details (e.g., hyper-parameters, model size) are missing from the paper.
> A: These details were described in Appendix B of our original submission because of the space limitation.
>
> Q: The results are average over only 3 seeds, is this enough to compare different algorithms?
> A: We have run all the SC2 experiments with more random seeds and found that the variances of learning curves are similar and the performance comparison results do not change. We have shown learning curves averaged over 5 different random seeds in Fig. 7, 8, and 10 in the updated version of our paper.
>
> Q: How beta should be determined?
> A: $\beta$ is used to trade off communication costs and communication effects. We have studied how $\beta$ affects the message embedding on the task sensor, as shown in Fig.3 on page 6. We also find that the performance of our method is robust across all the tested environments when $10^{-5} \le \beta \le 10^{-3}$. Therefore, we recommend that a $\beta$ in this region being tried first on new tasks and some fine-tuning may improve the performance further.
>
> Q: The authors present results when 80% of the messages are cut off. What is the performance of the model when all communication is cut off for comparison?
> A: We have shown the performance comparison when all communication is cut off, which is illustrated by Fig. 10 on page 16 of our original submission.
>
> Q: How does the approach work in competitive environments?
> A: Our approach is designed for a team of agents to learn to effectively collaborate and coordinate. These cooperative scenarios are common in the field of MARL [Foerster et al., AAAI 2018, Rashid et al., ICML 2018]. Of course, such a team of agents can compete against another opponent team in competitive settings.
>
> In mixed cooperative-competitive tasks, our method is readily combined with IC3Net [Singh et al., ICLR 2019], learning gates to cut off messages sent to the opponents. It will be the same as how TarMAC [Das et al., ICML 2019] is adapted to mixed environments. However, this setting is not the focus of this paper, and we will explore it in the future.
>
> Q: The experimental results section is not well organized. The authors mention five questions on page 6, but it is not very clear which examples/set of experiments address which question.
> A: Performance comparison and visualization results of didactic experiments can clarify all the questions. Our SC2 experiments further prove that the miscoordination problem of full decomposition methods is quite common and that our method can outperform QMIX and a state-of-the-art attentional communication algorithm.
>
> We hope that our clarifications can address your questions. Please let us know if you have any other questions.
>
> [Foerster et al., AAAI 2018] Foerster, J.N., Farquhar, G., Afouras, T., Nardelli, N. and Whiteson, S., 2018, April. Counterfactual multi-agent policy gradients. In Thirty-Second AAAI Conference on Artificial Intelligence.
>
> [Rashid et al., ICML 2019] Rashid, T., Samvelyan, M., Witt, C.S., Farquhar, G., Foerster, J. and Whiteson, S., 2018, July. QMIX: Monotonic Value Function Factorisation for Deep Multi-Agent Reinforcement Learning. In International Conference on Machine Learning (pp. 4292-4301).
>
> [Singh et al., ICLR 2019] Singh, A., Jain, T. and Sukhbaatar, S., 2019. Learning when to communicate at scale in multiagent cooperative and competitive tasks. In International Conference on Learning Representations.
>
> [Das et al., ICML 2019] Das, A., Gervet, T., Romoff, J., Batra, D., Parikh, D., Rabbat, M. and Pineau, J., 2019, May. TarMAC: Targeted Multi-Agent Communication. In International Conference on Machine Learning (pp. 1538-1546).

---

### Official Review · AnonReviewer1 · 2019-10-25
**Official Blind Review #1**

**Rating:** 6

**Review:**

Authors propose a new method for multi-agent reinforcement learning by using nearly decomposable value functions. The main idea is to have local (agent specific) value function and one global value function (for all agents). They also try minimizing the required communication need for multi-agent setup. To this end they deploy variational inference tools and support their method with an experimental study.

I found the paper interesting and empirical evaluation is good. However, my knowledge in the field is quite limited.


**Experience Assessment:**

I do not know much about this area.

**Review Assessment: Checking Correctness Of Derivations And Theory:**

I assessed the sensibility of the derivations and theory.

**Review Assessment: Checking Correctness Of Experiments:**

I assessed the sensibility of the experiments.

**Review Assessment: Thoroughness In Paper Reading:**

I made a quick assessment of this paper.

---

> ### Author Response · Authors · 2019-11-11
> **Thanks for your comments and interest in our paper.**
>
> Thanks for your comments and interest in our paper.
>
> Recently, the learning paradigm of value function factorization stands out as an effective approach for multi-agent deep reinforcement learning. [Samvelyan et al., AAMAS 2019] shows that its performance is much better than multi-agent policy gradients and independent value-based algorithm on StarCraft II micromanagement tasks.
>
> However, current value function factorization methods *fully* decompose the Q-functions among agents, and, as a result, each agent acts solely on its own local observations during execution. This full decomposition is not effective in many cases, because it is quite common that an agent’s optimal decision makings sometimes depend on information from other agents.
>
> To address these limitations, in this paper, we develop a novel function factorization method that learns nearly decomposable value functions. Our model learns not only Q-functions for agents but also communication protocols for their improved coordination. Different from other communication approaches, we explicitly formalize novel optimization objectives for minimizing communication so that only useful and necessary information (those can reduce uncertainty in value functions of other agents) can be exchanged between agents. These novelties explain why our method can significantly improve the performance of QMIX on StarCraft II micromanagement tasks even when 80% of the messages are cut off while other communication methods, e.g., TarMAC [Das et al., ICML 2019], can not.
>
> [Samvelyan et al., AAMAS 2019] Samvelyan, M., Rashid, T., Schroeder de Witt, C., Farquhar, G., Nardelli, N., Rudner, T.G., Hung, C.M., Torr, P.H., Foerster, J. and Whiteson, S., 2019, May. The starcraft multi-agent challenge. In Proceedings of the 18th International Conference on Autonomous Agents and Multi-Agent Systems (pp. 2186-2188). International Foundation for Autonomous Agents and Multi-agent Systems.
>
> [Das et al., ICML 2019] Das, A., Gervet, T., Romoff, J., Batra, D., Parikh, D., Rabbat, M. and Pineau, J., 2019, May. TarMAC: Targeted Multi-Agent Communication. In International Conference on Machine Learning (pp. 1538-1546).

---

### Official Review · AnonReviewer4 · 2019-11-01
**Official Blind Review #4**

**Rating:** 6

**Review:**

The paper tackles the collaborative multi-agent RL problem as the problem of finding almost-decentralized value functions, where the loss tries to minimize communications between the agents. The core idea is around maximizing the mutual information between each massage and the existing knowledge of its receiver. This way, redundant messages are naturally removed. The authors then assert an entropy regularization to (almost) prevent the agents from *cheating*. The paper is in general well-written and motivated. There are however certain issues that should be addressed. [The second one is my main issue.]

1. The term *message* has been used repeatedly but not defined. You should define precisely what you mean by a message in the background section. In particular, it is not clear from the text what a message looks like mathematically. You may also consider giving intuitive examples of how different ways of designing a message alters the behaviour in a given context before start talking about how to optimise them.

2. Section 3.1 needs a revisit. I assume by "optimal action policy of agent j" you mean "optimal policy of agent j". Formally, an optimal policy is greedy to the optimal value function and is deterministic unless there exist multiple actions with same optimal value at a given state. Therefore, the mutual information is not mathematically well-defined since most of the time the optimal policy is deterministic and does not induce a probability distribution.

2.1 What is the optimal value function of an agent? The agents are not optimizing their own value function, so even if your complex model converges, it does not imply optimality of agent-level value functions (and they shouldn’t be locally optimal). If by that you mean the agent-level value functions *after* convergence of $Q_tot$, then you need to clarify it to avoid the confusion. Even so, it is still not clear how you conceive the agent-level policies from these value functions.

2.2. Minor: As for the notation, I found $A_j$ quite strange for a policy; you may want to consider something like $\pi_j$.

3. If $f_m$ is learnt, then message encoding is not stationary at least in the training time. It may make the training potentially become unstable. Specifically, there is no condition to assure stability of your model. Suggestion: your entropy regularization might be sufficient to induce stability. More discussion on this (or a simple experiment to show if an instability exists and will be alleviated with regularization) would be quite helpful.

4. If I understand it correctly [and it wasn't clear from the text], the encoder $f_m$ is conditioned on local history, which induces that all the agents must share same history shape (tensor-wise), which in turn means that all the local agents has to have same local state shape (computationally, they should for example have same output/internal-state shape in their neural networks). This sounds like a limitation, and it may only be OK in domains where agents are homogenous.


**Experience Assessment:**

I do not know much about this area.

**Review Assessment: Checking Correctness Of Derivations And Theory:**

I assessed the sensibility of the derivations and theory.

**Review Assessment: Checking Correctness Of Experiments:**

I did not assess the experiments.

**Review Assessment: Thoroughness In Paper Reading:**

I read the paper at least twice and used my best judgement in assessing the paper.

---

> ### Author Response · Authors · 2019-11-11
> **Thanks for your detailed and constructive comments. Here we provide explanations to clarify your questions.**
>
> Thanks for your detailed and constructive comments. Here we provide explanations to clarify your questions.
>
> Q1: Precise definition of the message.
> A1: A message $m_{ij}$ from agent $i$ to agent $j$ encodes necessary information of the local observation-action history of agent $i$, which is helpful for the decision making of agent $j$. Precisely, a message $m_{ij}$ is defined as a sample drawn from a multivariate Gaussian distribution, whose covariance is a unit matrix and whose mean is given by an encoder $f_m(\tau_i, j;\theta_c)$, where $\tau_i$ is the local observation-action history of agent $i$ and $\theta_c$ are parameters of the encoder $f_m$. As the reviewer suggested, we have refined and moved this definition to the beginning of Sec. 3 in the updated version of our paper.
>
> Q2: Confusion about "optimal action policy of agent $j$" in Sec 3.1.
> A2: Sorry, this is a typo. It should be “action selection of agent $j$”.  As shown in Fig. 1, the mutual information is defined between messages and action selection. The variable $A_j$ used in the definition of mutual information is a random variable, whose probability distribution describes how the actions are selected by agent $j$. As action selection may depend on other agents, action selection can be stochastic and thus the mutual information is still well-defined. We have added more descriptions in the updated version of our paper.
>
> Q3: Stability issue of our model.
> A3: Like QMIX, during the training, our model is just one neural network (the message encoder can be regarded as some hidden layers), which is trained in an end-to-end fashion. Therefore, the message encoder will not introduce extra instability, and our model is faced with similar stability issue with QMIX. As in QMIX (and many other works in deep reinforcement learning), we use target networks, for both value functions and the message encoder, to stabilize training.
>
> Q4: The same history shape of agents limits the domains that our method can be applied.
> A4: Like most algorithms in MARL, we share network structure and parameters among agents. Such sharing is feasible because in common MARL benchmarking environments, such as SC2 unit micromanagement [Samvelyan et al., AAMAS 2019] and MPE [Lowe et al., NeurIPS 2017], heterogeneous agents have observation and action spaces with the same shape. In our SC2 experiments, results on three tasks (1o2r_vs_4r, 1o10b_vs_1r, and MMM) verify that our approach works well with heterogeneous agents.
>
> We hope that our clarification can address your questions. Please let us know if you have any other questions.
>
> [Samvelyan et al., AAMAS 2019] Samvelyan, M., Rashid, T., Schroeder de Witt, C., Farquhar, G., Nardelli, N., Rudner, T.G., Hung, C.M., Torr, P.H., Foerster, J. and Whiteson, S., 2019, May. The starcraft multi-agent challenge. In Proceedings of the 18th International Conference on Autonomous Agents and MultiAgent Systems (pp. 2186-2188). International Foundation for Autonomous Agents and Multiagent Systems.
>
> [Lowe et al., NeurIPS 2017] Lowe, R., Wu, Y., Tamar, A., Harb, J., Abbeel, O.P. and Mordatch, I., 2017. Multi-agent actor-critic for mixed cooperative-competitive environments. In Advances in Neural Information Processing Systems (pp. 6379-6390).

---

> > ### Comment · AnonReviewer4 · 2019-11-15
> > **Final**
> >
> > Thank you for your rebuttal and for partially addressing my comments. Point 2 and 2.1 are still in place. My assessment remains unchanged.

---

> > > ### Author Response · Authors · 2019-11-15
> > > **Clarifications of point 2 and 2.1**
> > >
> > > Thank you for your feedback.
> > >
> > > For point 2:
> > > In our formulation, we use the action selection variable $A_j$ of agent $j$, which is deterministic only when the histories of *all* agents are given. However, in the definition of the mutual information $I(A_j; M_{ij} | \mathrm{T}_j, M_{(-i)j})$, the history of agent $i$ *is not given*, and the action selection of agent $j$ is stochastic for its local history random variable $\mathrm{T}_j$ and the message random variables $M_{(-i)j}$. Thus, $A_j$ is a random variable.
> > >
> > > For point 2.1:
> > > In the value function factorization framework, optimal value functions of agents satisfy the Individual-Global-MAX (IGM) property:
> > >
> > > $$ argmax_{\mathbf{a}} Q_{tot}(\mathbf{\tau}, \mathbf{a})=
> > > \mathit{<}argmax_{a_1} Q_1(\tau_1, a_1), argmax_{a_2} Q_2(\tau_2, a_2), \dots, argmax_{a_n} Q_n(\tau_n, a_n) \mathit{>}
> > > $$
> > > (please refer to Eq.4 in QMIX [1] and Eq. 1 in QTRAN [2]). All value function factorization methods fulfill this property (QMIX [1], QTRAN [2], and VDN [3]). Our framework is based on QMIX, so the IGM property is guaranteed by the monotonic mixing network.
> > >
> > >
> > >
> > >
> > > [1] Rashid, T., Samvelyan, M., Witt, C.S., Farquhar, G., Foerster, J. and Whiteson, S., 2018, July. QMIX: Monotonic Value Function Factorisation for Deep Multi-Agent Reinforcement Learning. In International Conference on Machine Learning (pp. 4292-4301).
> > >
> > > [2] Son, K., Kim, D., Kang, W.J., Hostallero, D.E. and Yi, Y., 2019, May. QTRAN: Learning to Factorize with Transformation for Cooperative Multi-Agent Reinforcement Learning. In International Conference on Machine Learning (pp. 5887-5896).
> > >
> > > [3] Sunehag, P., Lever, G., Gruslys, A., Czarnecki, W.M., Zambaldi, V., Jaderberg, M., Lanctot, M., Sonnerat, N., Leibo, J.Z., Tuyls, K. and Graepel, T., 2018, July. Value-decomposition networks for cooperative multi-agent learning based on team reward. In Proceedings of the 17th International Conference on Autonomous Agents and MultiAgent Systems (pp. 2085-2087).

---

### Author Response · Authors · 2019-11-11
**Manuscript updated**

Dear reviewers,

We have updated the manuscript to improve its presentation based on your comments. Thank you again for your feedback.

---

### Decision · Program_Chairs · 2019-12-19

**Decision:**

Accept (Poster)

**Comment:**

The paper extends recent value function factorization methods for the case where limited agent communication is allowed. The work is interesting and well motivated. The reviewers brought up a number of mostly minor issues, such as unclear terms and missing implementation details. As far as I can see, the reviewers have addressed these issues successfully in their updated version. Hence, my recommendation is accept.